# Rapid, Reference-Free human genotype imputation with denoising autoencoders

Raquel Dias[1,2,3], Doug Evans[1,2], Shang-Fu Chen[1,2], Kai-Yu Chen[1,2], Salvatore Loguercio[1,2], Leslie Chan[1,2], Ali Torkamani[1,2]*

[1]Scripps Research Translational Institute, Scripps Research Institute, La Jolla, United States; [2]Department of Integrative Structural and Computational Biology, Scripps Research, La Jolla, United States; [3]Department of Microbiology and Cell Science, University of Florida, Gainesville, United States

**Abstract** Genotype imputation is a foundational tool for population genetics. Standard statistical imputation approaches rely on the co-location of large whole-genome sequencing-based reference panels, powerful computing environments, and potentially sensitive genetic study data. This results in computational resource and privacy-risk barriers to access to cutting-edge imputation techniques. Moreover, the accuracy of current statistical approaches is known to degrade in regions of low and complex linkage disequilibrium. Artificial neural network-based imputation approaches may overcome these limitations by encoding complex genotype relationships in easily portable inference models. Here, we demonstrate an autoencoder-based approach for genotype imputation, using a large, commonly used reference panel, and spanning the entirety of human chromosome 22. Our autoencoder-based genotype imputation strategy achieved superior imputation accuracy across the allele-frequency spectrum and across genomes of diverse ancestry, while delivering at least fourfold faster inference run time relative to standard imputation tools.

*For correspondence:
atorkama@scripps.edu

**Competing interest:** The authors declare that no competing interests exist.

## Editor's evaluation

The paper describes a novel neural-network-based strategy for imputing unmeasured genotypes, which is a standard part of most association testing pipelines. The method is computationally intensive to train, but once training is complete the imputation is fast and accurate and does not require further access to a reference panel. It has the potential to be a practically-appealing alternative to existing methods. although further work (eg training of models) is required before this new approach can be applied genome-wide.

## Introduction

The human genome is inherited in large blocks from parental genomes, generated through a DNA-sequence-dependent shuffling process called recombination. The non-uniform nature of recombination breakpoints producing these genomic blocks results in correlative genotype relationships across genetic variants, known as linkage disequilibrium. Thus, genotypes for a small subset (1–10%) of observed common genetic variants can be used to infer the genotype status of unobserved but known genetic variation sites across the genome (on the order of ~1 M of >10 M sites; *Li et al., 2009*; *Marchini and Howie, 2010*). This process, called genotype imputation, allows for the generation of nearly the full complement of known common genetic variation at a fraction of the cost of direct genotyping or sequencing. Given the massive scale of genotyping required for genome-wide association studies or implementation of genetically informed population health initiatives, genotype imputation is an essential approach in population genetics.

Standard approaches to genotype imputation utilize Hidden Markov Models (HMM) (**Browning et al., 2018**; **Das et al., 2016**; **Rubinacci et al., 2020**) distributed alongside large WGS-based reference panels (**Browning and Browning, 2016**). In general terms, these imputation algorithms use genetic variants shared between to-be-imputed genomes and the reference panel and apply Hidden Markov Models (HMM) to impute the missing genotypes per sample (**Das et al., 2018**). The hidden states in the HMMs represent the haplotype in a reference panel that is most closely related to the haplotype being imputed. The HMM parameter estimation also depends on recombination rates, mutation rates, and/or genotype error rates that must be fit by Markov Chain Monte Carlo Algorithm (MCMC) or an expectation-maximization algorithm. Thus, HMM-based imputation is a computationally intensive process, requiring access to both high-performance computing environments and large, privacy-sensitive, WGS reference panels (**Kowalski et al., 2019**). Often, investigators outside of large consortia will resort to submitting genotype data to imputation servers (**Das et al., 2016**), resulting in privacy and scalability concerns (**Sarkar et al., 2021**).

Recently, artificial neural networks, especially autoencoders, have attracted attention in functional genomics for their ability to fill-in missing data for image restoration and inpainting (**Chaitanya et al., 2017**; **Ghosh et al., 2020**; **Mao et al., 2016**; **Xie et al., 2012**). Autoencoders are neural networks tasked with the problem of simply reconstructing the original input data, with constraints applied to the network architecture or transformations applied to the input data in order to achieve a desired goal like dimensionality reduction or compression, and de-noising or de-masking (**Abouzid et al., 2019**; **Liu et al., 2020**; **Voulodimos et al., 2018**). Stochastic noise or masking is used to modify or remove data inputs, training the autoencoder to reconstruct the original uncorrupted data from corrupted inputs (**Tian et al., 2020**). Autoencoders that receive corrupted or masked data as input and are trained to predict the original uncorrupted data as the output are also known as denoising autoencoders. These autoencoder characteristics are well-suited for genotype imputation and may address some of the limitations of HMM-based imputation by eliminating the need for dissemination of reference panels and allowing the capture of

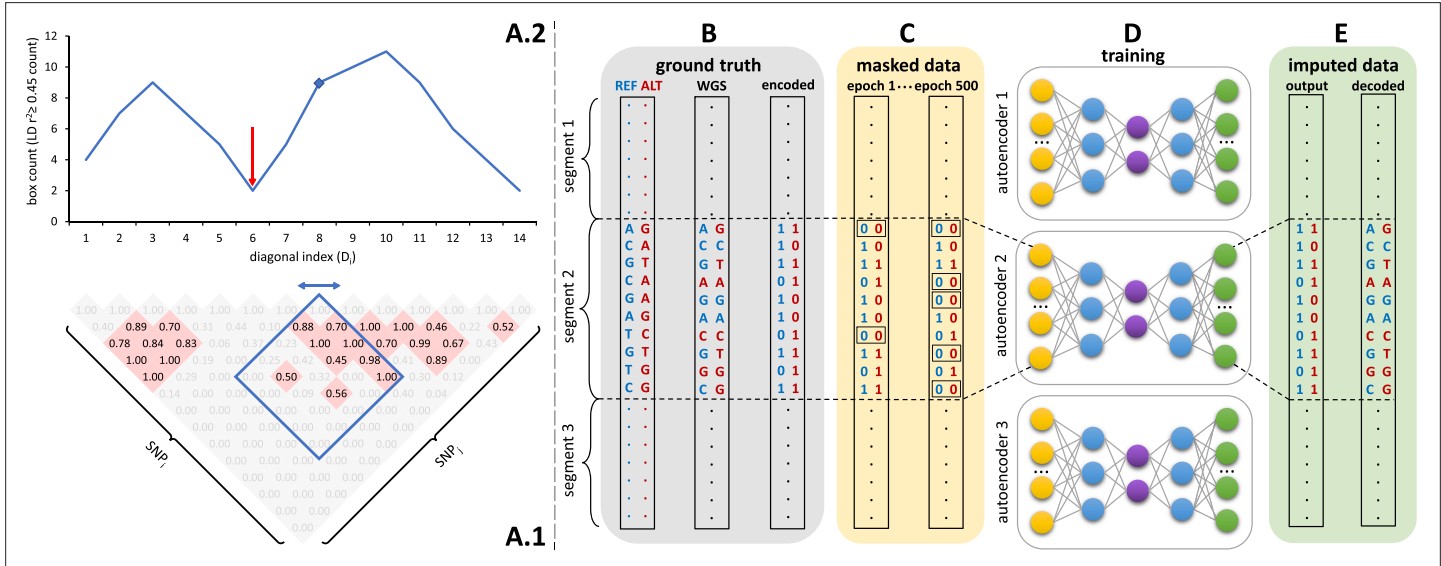

**Figure 1.** Schematic overview of the autoencoder training workflow. (**A**) Tiling of autoencoders across the genome is achieved by (**A.1**) calculating a n x n matrix of pairwise SNP correlations, thresholding them at 0.45 (selected values are shown in red background, excluded values in gray), (**A.2**) quantifying the overall local LD strength centered at each SNP by computing their local correlation box counts and splitting the genome into approximately independent segments by identifying local minima (recombination hotspots). The red arrow illustrates minima between strong LD regions. For reducing computational complexity, we calculated the correlations in a fixed sliding box size of 500x500 common variants (MAF ≥ 0.5%). Thus, the memory utilization for calculating correlations will be the same regardless of genomic density. (**B**) Ground truth whole genome sequencing data is encoded as binary values representing the presence (1) or absence (0) of the reference allele (blue) and alternative allele (red). (**C**) Variant masking (setting both alleles as absent, represented by 0, corrupts data inputs at a gradually increasing masking rate). Example masked variants are outlined. (**D**) Fully-connected autoencoders spanning segments defined as shown in panel (**A**), are then trained to reconstruct the original uncorrupted data from corrupted inputs; (**E**) the reconstructed outputs (imputed data) are compared to the ground truth states for loss calculation and are decoded back to genotypes.

non-linear relationships in genomic regions with complex linkage disequilibrium structures. Some attempts at genotype imputation using neural networks have been previously reported, though for specific genomic contexts (*Naito et al., 2021*) at genotype masking levels (5–20%) not applicable in typical real-world population genetics scenarios (*Chen and Shi, 2019*; *Islam et al., 2021*; *Sun and Kardia, 2008*), or in contexts where the neural network must be trained for imputation with specific input variant sets, i.e. the model must be re-trained for imputation from different geno-typing array (*Kojima et al., 2020*).

Here, we present a generalized approach to unphased human genotype imputation using sparse, denoising autoencoders capable of highly accurate genotype imputation at genotype masking levels (98+%) appropriate for array-based genotyping and low-pass sequencing-based population genetics initiatives. We describe the initial training and implementation of autoencoders spanning all of human chromosome 22, achieving equivalent to superior accuracy relative to modern HMM-based methods, and dramatically improving computational efficiency at deployment without the need to distribute reference panels.

## Materials and methods

### Overview

Sparse, de-noising autoencoders spanning all bi-allelic SNPs observed in the Haplotype Reference Consortium were developed and optimized. Each autoencoder receives masked data as input and is trained to predict the original uncorrupted data as the output. Each bi-allelic SNP was encoded as two binary input nodes, representing the presence or absence of each allele (*Figure 1B, E*). This encoding allows for the straightforward extension to multi-allelic architectures and non-binary allele presence probabilities. A data augmentation approach using modeled recombination events and offspring formation coupled with random masking at an escalating rate drove our autoencoder training strategy (*Figure 1C*). Because of the extreme skew of the allele frequency distribution for rarely present alleles (*Auton et al., 2015*), a focal-loss-based approach was essential to genotype imputation performance. The basic architecture of the template fully-connected autoencoder before optimization to each genomic segment is depicted in *Figure 1D*. Individual autoencoders were designed to span genomic segments with boundaries defined by computationally identified recombination hotspots (*Figure 1A*). The starting point for model hyperparameters were randomly selected from a grid of possible combinations and were further tuned from a battery of features describing the complexity of the linkage-disequilibrium structure of each genomic segment.

### Genotype encoding

Genotypes for all bi-allelic SNPs were converted to binary values representing the presence (1) or absence (0) of the reference allele A and alternative allele B, respectively, as shown in *Equation 1*.

$$
x_i = \begin{cases}
if\left(G_i = \left[A, A\right]\right): & x_i = \left[1, 0\right] \\
if\left(G_i = \left[A, B\right]\right): & x_i = \left[1, 1\right] \\
if\left(G_i = \left[B, A\right]\right): & x_i = \left[1, 1\right] \\
if\left(G_i = \left[B, B\right]\right): & x_i = \left[0, 1\right] \\
if\left(G_i = \left[null\right]\right): & x_i = \left[0, 0\right]
\end{cases}
\tag{1}
$$

where *x* is a vector containing the two allele presence input nodes to the autoencoder and their encoded allele presence values derived from the original genotype, *G*, of variant *i*. The output nodes of the autoencoder are similarly rescaled to 0–1 by a sigmoid function, split into three genotype outputs (homozygous reference, homozygous alternate, and heterozygous), and normalized using the Softmax function. The normalized outputs can also be regarded as probabilities and can be combined for the calculation of alternative allele dosage and as a measure of imputation quality. This representation is extensible to other classes of genetic variation, and allows for the use of probabilistic loss functions.

## Training data, masking, and data augmentation

### Training data

Whole-genome sequence data from the Haplotype Reference Consortium (HRC) was used for training and as the reference panel for comparison to HMM-based imputation (*McCarthy et al., 2016*). The dataset consists of 27,165 samples and 39,235,157 biallelic SNPs generated using whole-genome sequence data from 20 studies of predominantly European ancestry (HRC Release 1.1): 83.92% European, 2.33% East Asian, 1.63% Native American, 2.17% South Asian, 2.96% African, and 6.99% admixed ancestry individuals. Genetic ancestry was determined using continental population classification from the 1000 Genomes Phase3 v5 (1000 G) reference panel and a 95% cutoff using Admixture software (*Alexander et al., 2009*). Genotype imputation autoencoders were trained for all 510,442 unique SNPs observed in HRC on human chromosome 22. For additional comparisons, whole-genome sequence data from 31 studies available through the NHLBI Trans-Omics for Precision Medicine (TOPMed) program were used as an alternative reference panel for HMM-based imputation tools (*Taliun et al., 2021*). We downloaded Freeze 8 of TOPMed, which is the latest version with all consent groups genotyped across the same set of jointly called variants. GRCh38 TOPMed cohorts were converted to hg19 with Picard 2.25 ('Picard toolkit', 2019), and multi allelic SNPs removed with bcftools v.1.10.2 (*Danecek et al., 2021*). Any variants with missing genotypes were excluded as well, yielding a final reference panel for chr22 consisting of 73,586 samples and 11,089,826 biallelic SNPs. Since the ARIC and MESA cohorts are used for model selection and validation, they were excluded from the TOPMed reference panel. Relatedness analysis using the KING robust kinship estimator revealed significant data leakage boosting HMM-based imputation performance through individuals directly participating in the MESA and other TOPMed cohorts, as well as through numerous first- and second-degree familial relationships spanning MESA individuals and individuals in other TOPMed cohorts.

### Validation and testing data

A balanced (50%:50% European and African genetic ancestry) subset of 796 whole genome sequences from the Atherosclerosis Risk in Communities cohort (ARIC) (*Mou et al., 2018*), was used for model validation and selection. The Wellderly (*Erikson et al., 2016*), Human Genome Diversity Panel (HGDP) (*Cann et al., 2002*), and Multi-Ethnic Study of Atherosclerosis (MESA) (*Bild et al., 2002* ) cohorts were used for model testing. The Wellderly cohort consisted of 961 whole genomes of predominantly European genetic ancestry. HGDP consisted of 929 individuals across multiple ancestries: 11.84% European, 14.64% East Asian, 6.57% Native American, 10.98% African, and 55.97% admixed. MESA consisted of 5370 whole genomes across multiple ancestries: 27.62% European, 11.25% East Asian, 4.99% Native American, 5.53% African, and 50.61% admixed. MESA, Wellderly, and HGDP are all independent datasets, not used for autoencoder training, nor model selection, whereas HRC and ARIC were utilized for training and model selection, respectively.

GRCh38 mapped cohorts (HGDP and MESA) were converted to hg19 using Picard v2.25 (*Broad Institute, 2022*). All other datasets were originally mapped and called against hg19. Multi-allelic SNPs, SNPS with >10% missingness, and SNPs not observed in HRC were removed with bcftools v1.10.2 (*Danecek et al., 2021*). Mock genotype array data was generated from these WGS cohorts by restricting genotypes to those present on commonly used genotyping arrays (Affymetrix 6.0, UKB Axiom, and Omni 1.5 M). For chromosome 22, intersection with HRC and this array-like masking respectively resulted in: 9025, 10,615, and 14,453 out of 306,812 SNPs observed in ARIC; 8630, 10,325, and 12,969 out of 195,148 SNPs observed in the Wellderly; 10,176, 11,086, and 14,693 out of 341,819 SNPs observed in HGDP; 9237, 10,428, and 13,677 out of 445,839 SNPs observed in MESA. All input genotypes from all datasets utilized in this work are unphased, and no pre-phasing was performed.

### Data augmentation

We employed two strategies for data augmentation – random variant masking and simulating further recombination with offspring formation. During training, random masking of input genotypes was performed at escalating rates, starting with a relatively low masking rate (80% of variants) that is gradually incremented in subsequent training rounds until up to only five variants remain unmasked per autoencoder. Masked variants are encoded as the *null* case in *Equation 1*. During finetuning we

used sim1000G (*Dimitromanolakis et al., 2019*) to simulate of offspring formation using the default genetic map and HRC genomes as parents. A total of 30,000 offspring genomes were generated and merged with the original HRC dataset, for a total of 57,165 genomes.

## Loss function

In order to account for the overwhelming abundance of rare variants, the accuracy of allele presence reconstruction was scored using an adapted version of focal loss (*FL*) (*Lin et al., 2017*), shown in *Equation 2*.

$$FL = -\alpha_t \left(1 - p_t\right)^{\gamma} \left[x_t \log\left(p_t\right) + \left(1 - x_t\right) \log\left(1 - p_t\right)\right] \tag{2}$$

where the classic cross entropy (shown as binary log loss in brackets) of the truth class ($x_t$) predicted probability ($p_t$) is weighted by the class imbalance factor $\alpha_t$ and a modulating factor $(1 - p_t)^{\gamma}$. $t$ represents the index of each allele in a genomic segment. The modulating factor is the standard focal loss factor with hyperparameter, $\gamma$, which amplifies the focal loss effect by down-weighting the contributions of well-classified alleles to the overall loss (especially abundant reference alleles for rare variant sites)(*Lin et al., 2017*). $\alpha_t$ is an additional balancing hyperparameter set to the truth class frequency.

This base focal loss function is further penalized and regularized to encourage simple and sparse models in terms of edge-weight and hidden layer activation complexity. These additional penalties result in our final loss function as shown in *Equation 3*.

$$SFL = -\alpha_t \left(1 - p_t\right)^{\gamma} \left[x_t \log\left(p_t\right) + \left(1 - x_t\right) \log\left(1 - p_t\right)\right] + \beta S_{\left(\rho\|\hat{\rho}\right)} + \lambda_1 \|W\|_1 + \lambda_2 \|W\|_2 \tag{3}$$

where $\|W\|_1$ and $\|W\|_2$ are the standard *L1* and *L2* norms of the autoencoder weight matrix (W), with their contributions mediated by the hyperparameters $\lambda_1$ and $\lambda_2$. S is a sparsity penalty, with its contribution mediated by the hyperparameter $\beta$, which penalizes deviation from a target hidden node activation set by the hyperparameter vs the observed mean activation $\rho$ over a training batch $j$ summed over total batches $n$, as shown in *Equation 4*:

$$S_{\left(\rho\|\hat{\rho}\right)} = \sum_{j=1}^{n} \rho * log\left(\frac{\rho}{\hat{\rho}_j}\right) + \left(1 - \rho\right) * log\left(\frac{1-\rho}{1-\hat{\rho}_j}\right) \tag{4}$$

## Genome tiling

All model training tasks were distributed across a diversified set of NVIDIA graphical processing units (GPUs) with different video memory limits: 5 x Titan Vs (12 GB), 8x A100s (40 GB), 60x V100s (32 GB). Given computational complexity and GPU memory limitations, individual autoencoders were designed to span approximately independent genomic segments with boundaries defined by computationally identified recombination hotspots (*Figure 1E*). These segments were defined using an adaptation of the LDetect algorithm (*Berisa and Pickrell, 2016*). First, we calculated a $n$ x $n$ matrix of pairwise SNP correlations using all common genetic variation (≥0.5% minor allele frequency) from HRC. Correlation values were thresholded at 0.45, which is the threshold that returns the minimum number of segments spanning chromosome 22 with an average size per segment that fits into the video memory of GPUs. While developing the tiling algorithm, we tested lower thresholds, which made the segments smaller and more abundant, and thus made the GPU memory workload less efficient (e.g. many tiles resulted in many autoencoders per GPU, which thus caused a CPU-GPU communication overhead). Due to the obstacles related to computational inefficiency, CPU-GPU communication overhangs, and GPU memory limits, we did not proceed with model training on segments generated with other correlation thresholds. For each SNP, we calculated a box count of all pairwise SNP correlations spanning 500 common SNPs upstream and downstream of the index SNP. This moving box count quantifies the overall local LD strength centered at each SNP. Local minima in this moving box count were used to split the genome into approximately independent genomic segments of two types – large segments of high LD interlaced with short segments of weak LD corresponding to recombination hotspot regions. Individual autoencoders were designed to span the entirety of a single high LD segment plus its adjacent upstream and downstream weak LD regions. Thus, adjacent autoencoders overlap at their weak LD ends. If an independent genomic segment exceeded the threshold number of SNPs amenable to deep learning given GPU memory limitations, internal local minima within the high LD regions were used to split the genomic segments further to a maximum of 6000 SNPs per

**Table 1.** Description and values of hyperparameters tested in grid search.
$\lambda_1$: scaling factor for Least Absolute Shrinkage and Selection Operator (LASSO or L1) regularization; $\lambda_2$: scaling factor for Ridge (L2) regularization; β: scaling factor for sparsity penalty described in *Equation 4*; $\rho$: target hidden layer activation described in *Equation 4*; Activation function type: defines how the output of a hidden neuron will be computed given a set of inputs; Learning rate: step size at each learning iteration while moving toward the minimum of the loss function; $\gamma$: amplifying factor for focal loss described in *Equation 3*; Optimizer type: algorithms utilized to minimize the loss function and update the model weights in backpropagation; Loss type: algorithms utilized to calculate the model error (*Equation 2*); Number of hidden layers: how many layers of artificial neurons to be implemented between input layer and output layer; Hidden layer size ratio: scaling factor to resize the next hidden layer with reference to the size of Its previous layer; Learning rate decay ratio: scaling factor for updating the learning rate value on every 500 epochs.

| Hyperparameter description | Tested values (coarse-grid search) |
| --- | --- |
| $\lambda_1$ for L1 regularization | [1e-3, 1e-4, 1e-5, 1e-6, 1e-1, 1e-2, 1e-7, 1e-8] |
| $\lambda_2$ for L2 regularization | [1e-3, 1e-4, 1e-5, 1e-6, 1e-1, 1e-2, 1e-7, 1e-8] |
| Sparsity scaling factor (β) | [0, 0.001, 0.01, 0.05, 1, 5, 10] |
| Target average hidden layer activation ($\rho$) | [0.001, 0.004, 0.007, 0.01, 0.04, 0.07, 0.1, 0.4, 0.7, 1.0] |
| Activation function type | ['sigmoid', 'tanh', 'relu', 'softplus'] |
| Learning rate | [0.000001, 0.00001, 0.0001, 0.001, 0.01, 0.1, 1, 10, 100] |
| Amplifying factor for focal loss (γ) | [0, 0.5, 1, 2, 3, 5] |
| Optimizer type | ['Adam', 'RMS Propagation', 'Gradient Descent'] |
| Loss type | ['Binary Cross Entropy', 'Custom Focal Loss'] |
| Number of hidden layers | [1, 2, 4, 6, 8] |
| Hidden layer size ratio | [0.2, 0.3, 0.4, 0.5, 0.6, 0.7, 0.8, 0.9, 1] |
| Learning rate decay ratio | [ 0.0, 0.25, 0.5, 0.75, 0.95, 0.99, 0.999, 0.9999] |

autoencoder. Any remaining genomic segments still exceeding 6000 SNPs were further split into 6000 SNP segments with large overlaps of 2500 SNPs given the high degree of informative LD split across these regions. This tiling process resulted in 256 genomic segments spanning chromosome 22: 188 independent LD segments, 32 high LD segments resulting from internal local minima splits, and 36 segments further split due to GPU memory limitations.

## Hyperparameter initialization and grid search

We first used a random grid search approach to define initial hyperparameter combinations producing generally accurate genotype imputation results. The hyperparameters and their potential starting values are listed in *Table 1*. This coarse-grain grid search was performed on all genomic segments of chromosome 22 (256 genomic segments), each tested with 100 randomly selected hyperparameter combinations per genomic segment, with a batch size of 256 samples, training for 500 epochs without any stop criteria, and validating on an independent dataset (ARIC). To evaluate the performance of each hyperparameter combination, we calculated the average coefficient of determination (r-squared) comparing the predicted and observed alternative allele dosages per variant. Concordance and F1-score were also calculated to screen for anomalies but were not ultimately used for model selection.

## Hyperparameter tuning

In order to avoid local optimal solutions and reduce the hyperparameter search space, we used an ensemble-based machine learning approach (Extreme Gradient Boosting–XGBoost) to predict the expected performance (r-squared) of each hyperparameter combination per genomic segment using the results of the coarse-grid search and predictive features calculated for each genomic segment. These features include the number of variants, average recombination rate and average pairwise

Pearson correlation across all SNPs, proportion of rare and common variants across multiple minor allele frequency (MAF) bins, number of principal components necessary to explain at least 90% of variance, and the total variance explained by the first two principal components. The observed accuracies of the coarse-grid search, numbering 25,600 training inputs, were used to predict the accuracy of 500,000 new hyperparameter combinations selected from *Table 1* without training. All categorical predictors (activation function name, optimizer type, loss function type) were one-hot encoded. The model was implemented using XGBoost package v1.4.1 in Python v3.8.3 with 10-fold cross-validation and default settings.

We then ranked all hyperparameter combinations by their predicted performance and selected the top 10 candidates per genomic segment along with the single best initially tested hyperparameter combination per genomic segments for further consideration. All other hyperparameter combinations were discarded. Genomic segments with sub-optimal performance relative to Minimac were subjected to tuning with simulated offspring formation. For tuning, the maximum number of epochs was increased (35,000) with automatic stop criteria: if there is no improvement in average loss value of the current masking/training cycle versus the previous one, the training is interrupted, otherwise training continues until the maximum epoch limit is reached. Each masking/training cycle consisted of 500 epochs. Final hyperparameter selection was based on performance on the validation dataset (ARIC).

This process results in 256 unique autoencoders spanning the genomic segments of chromosome 22. Each genomic segment consists of a different number of input variables (genetic variants), sparsity, and correlation structure. Thus, 256 unique autoencoder models span the entirety of chromosome 22 (e.g.: each autoencoder has different edge weights, number of layers, loss function, as well as regularization and optimization parameters).

## Performance testing and comparisons

Performance was compared to Minimac4 (*Das et al., 2016*), Beagle5 (*Browning et al., 2018*), and Impute5 (*Rubinacci et al., 2020*) using default parameters. We utilized HRC as reference panel for the HMM-based imputation tools, which is the same dataset used for training the autoencoders, and we applied the same quality control standards for both HMM-based and autoencoder-based imputation. We also provide additional comparisons to HMM-based imputation using the TOPMed cohort. No post-imputation quality control was applied. Population level reconstruction accuracy is quantified by measuring r-squared across multiple strata of data: per genomic segment, at whole chromosome level, and stratified across multiple minor allele frequency bins: [0.001–0.005], [0.005–0.01], [0.01–0.05], [0.05–0.1], [0.1–0.2], [0.2–0.3], [0.3–0.4], [0.4–0.5]. While r-squared is our primary comparison metric, sample-level and population-level model performance is also evaluated with concordance and the F1-score. Wilcoxon rank-sum testing was used to assess the significance of accuracy differences observed. Spearman correlations were used to evaluate the relationships between genomic segment features and observed imputation accuracy differences. Standard errors for per variant imputation accuracy r-squared is equal or less than 0.001 where not specified. Performance is reported only for the independent test datasets (Wellderly, MESA, and HGDP). Note that MESA ultimately is not independent of the TOPMed cohort when used for HMM-based imputation.

We used the MESA cohort for inference runtime comparisons. Runtime was determined using the average and standard error of three imputation replicates. Two hardware configurations were used for the tests: (1) a low-end environment: 16-core Intel Xeon CPU (E5-2640 v2 2.00 GHz), 250 GB RAM, and one GPU (NVIDIA GTX 1080); (2) a high-end environment: 24-Core AMD CPU (EPYC 7352 2.3 GHz), 250 GB RAM, using one NVIDIA A100 GPU. We report computation time only, input/output (I/O) reading/writing times are excluded as separately optimized functions. Since the computational burden of training the models remains on the developer side, the runtime results refer to the task of imputing the missing genotypes given a pre-trained autoencoder set.

## Data availability

The data that support the findings of this study are available from dbGAP and European Genome-phenome Archive (EGA), but restrictions apply to the availability of these data, which were used under ethics approval for the current study, and so are not openly available to the public. The computational pipeline for autoencoder training and validation is available at https://github.com/TorkamaniLab/

Imputation_Autoencoder/tree/master/autoencoder_tuning_pipeline; *Dias et al., 2022*. The python script for calculating imputation accuracy is available at https://github.com/TorkamaniLab/impu-tation_accuracy_calculator; *Dias, 2021*. Instructions on how to access the unique information on the parameters and hyperparameters of each one of the 256 autoencoders is shared through our source code repository at https://github.com/TorkamaniLab/imputator_inference, copy archived at swh:1:rev:2fbd203acf8aaf320a520c6374d6f4d57f068a7c; *Dias, 2022*. We also shared the pre-trained autoencoders and instructions on how to use them for imputation at https://github.com/Torkama-niLab/imputator_inference; *Dias, 2022*.

## Imputation data format

The imputation results are exported in variant calling format (VCF) containing the imputed genotypes and imputation quality scores in the form of class probabilities for each one of the three possible

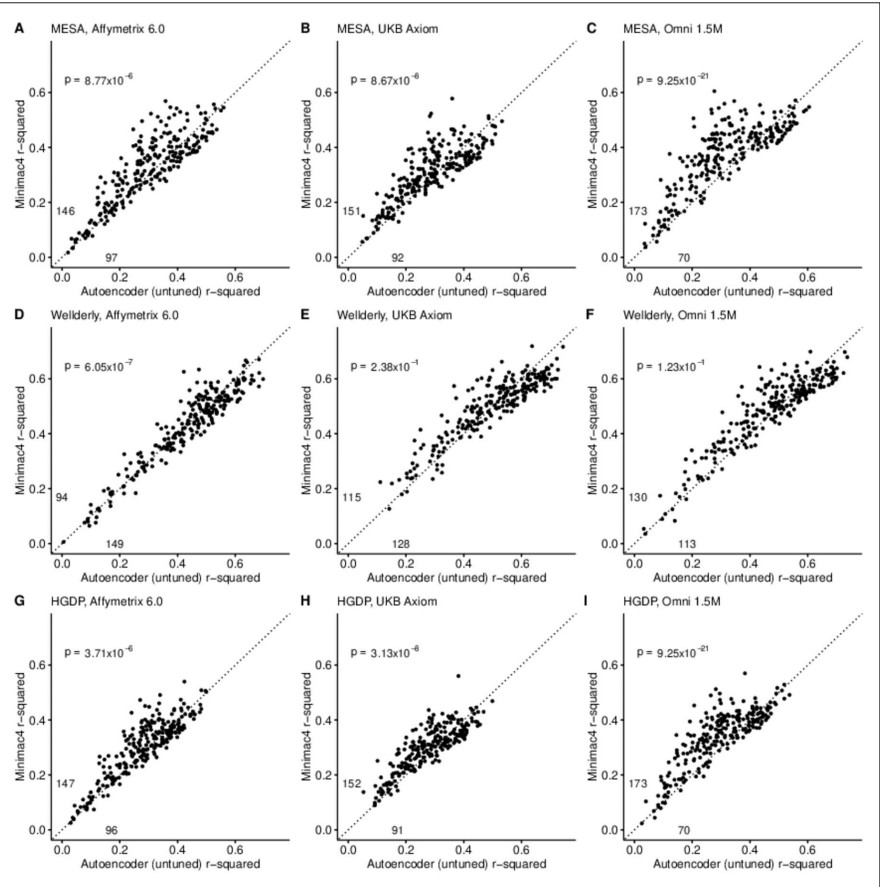

**Figure 2.** HMM-based (y-axis) versus autoencoder-based (x-axis) imputation accuracy prior to tuning. Minimac4 and untuned autoencoders were tested across three independent datasets–- MESA (top), Wellderly (**middle**), and HGDP (**bottom**) and across three genotyping array platforms–- Affymetrix 6.0 (**left**), UKB Axiom (**middle**), Omni1.5M (**right**). Each data point represents the imputation accuracy (average r-squared per variant) for an individual genomic segment relative to its WGS-based ground truth. The numerical values presented on the left side and below the identity line (dashed line) indicate the number of genomic segments in which Minimac4 outperformed the untuned autoencoder (left of identity line) and the number of genomic segments in which the untuned autoencoder surpassed Minimac4 (below the identity line). Statistical significance was assessed through two-proportion Z-test p-values.

The online version of this article includes the following figure supplement(s) for figure 2:

**Figure supplement 1.** Beagle5 (y-axis) versus autoencoder-based (x-axis) imputation accuracy prior to tuning.

**Figure supplement 2.** Impute5 (y-axis) versus autoencoder-based (x-axis) imputation accuracy prior to tuning.

**Figure supplement 3.** Relationship between genomic segment features and autoencoder performance.

**Figure supplement 4.** Projecting autoencoder performance from hyperparameters and genomic features.

**Table 2.** Performance comparisons between untuned autoencoder (AE) and HMM-based imputation tools (Minimac4, Beagle5, and Impute5).

Average r-squared per variant was extracted from each genomic segment of chromosome 22. We applied Wilcoxon rank-sum tests to compare the HMM-based tools to the reference tuned autoencoder (AE). * represents p-values ≤0.05, ** indicates p-values ≤0.001, and *** indicates p-values ≤0.0001.

| | MESA | Wellderly | HGDP | Affymetrix 6.0 | UKB Axiom | Omni 1.5 M | Combined |
|---|---|---|---|---|---|---|---|
| AE (untuned) | 0.303±0.008 | 0.470±0.009 | 0.285±0.006 | 0.339±0.008 | 0.356±0.007 | 0.362±0.008 | 0.352±0.008 |
| Minimac4 | 0.337±0.007* | 0.471±0.008 | 0.314±0.006** | 0.352±0.008 | 0.370±0.006 | 0.400±0.007** | 0.374±0.007* |
| Beagle5 | 0.336±0.007* | 0.460±0.008 | 0.296±0.005 | 0.342±0.007 | 0.367±0.006 | 0.384±0.007* | 0.364±0.007 |
| Impute5 | 0.326±0.007* | 0.458±0.008 | 0.289±0.006 | 0.336±0.008 | 0.354±0.006 | 0.383±0.008* | 0.358±0.007 |

genotypes (homozygous reference, heterozygous, and homozygous alternate allele). The probabilities can be used for quality control of the imputation results.

## Results

### Untuned performance and model optimization

A preliminary comparison of the best performing autoencoder per genomic segment vs HMM-based imputation was made after the initial grid search (Minimac4: *Figure 2*, Beagle5 and Eagle5: *Figure 2—figure supplements 1–2*). Untuned autoencoder performance was generally inferior to all tested HMM-based methods except when tested on the European ancestry-rich Wellderly dataset when masked using the Affymetrix 6.0 and UKB Axiom marker sets, but not Omni 1.5 M markers. HMM-based imputation was consistently superior across the more ancestrally diverse test datasets (MESA and HGDP) (two proportion test, $p \leq 8.77 \times 10^{-6}$). Overall, when performance across genomic segments, test datasets, and test array marker sets was combined, the autoencoders exhibited an average r-squared per variant of 0.352±0.008 in reconstruction of WGS ground truth genotypes versus an average r-squared per variant of 0.374±0.007, 0.364±0.007, and 0.357±0.007 for HMM-based imputation methods (Minimac4, Beagle5, and Impute5, respectively) (*Table 2*). This difference was statistically significant only relative to Minimac4 (Minimac4: Wilcoxon rank-sum test p=0.037, Beagle5 and Eagle5: p≥0.66).

In order to understand the relationship between genomic segment features, hyperparameter values, and imputation performance, we calculated predictive features (see Materials and methods) for

**Table 3.** Top 10 best performing hyperparameter combinations that advanced to fine-tuning.
See Materials and methods and *Table 1* for a detailed description of the hyperparameters.

| $\lambda_1$ | $\lambda_2$ | $\beta$ | $\rho$ | Activation | Learn rate | $\gamma$ | Optimizer | Loss type | Hidden layers | Size ratio | Decay |
|---|---|---|---|---|---|---|---|---|---|---|---|
| 0.1 | 0 | 0.01 | 0.01 | tanh | $1.0*10^{-4}$ | 0 | adam | CE | 4 | 1 | 0.95 |
| 0.1 | 0 | 1 | 0.5 | sigmoid | $1.0*10^{-4}$ | 1 | adam | CE | 2 | 0.9 | 0.95 |
| 0.1 | 0 | 5 | 0.5 | sigmoid | $1.0*10^{-1}$ | 4 | adam | CE | 2 | 0.5 | 0 |
| 0.1 | 0 | 1 | 0.005 | relu | $1.0*10^{-1}$ | 4 | adam | FL | 6 | 1 | 0.25 |
| 0.1 | 0 | 5 | 0.01 | relu | $1.0*10^{-5}$ | 5 | adam | FL | 4 | 1 | 0.95 |
| 0.1 | 0 | 0.01 | 0.1 | leakyrelu | $1.0*10^{-5}$ | 0 | adam | FL | 8 | 0.9 | 0.95 |
| 0.1 | 0 | 1 | 0.01 | tanh | $1.0*10^{-4}$ | 0 | adam | CE | 6 | 1 | 0.95 |
| 0 | $1.0*10^{-8}$ | 0.001 | 0.05 | relu | $1.0*10^{-5}$ | 4 | adam | CE | 8 | 0.6 | 0.95 |
| 0.1 | 0 | 0 | 0.01 | relu | $1.0*10^{-1}$ | 5 | adam | FL | 8 | 0.9 | 0 |
| 0.1 | 0 | 0.01 | 0.01 | tanh | $1.0*10^{-3}$ | 5 | adam | CE | 2 | 1 | 0.95 |

each genomic segment and determined their Spearman correlation with the differences in r-squared observed for the autoencoder vs Minimac4 (*Figure 2—figure supplement 3*). We observed that the autoencoder had superior performance when applied to the genomic segments with the most complex LD structures: those with larger numbers of observed unique haplotypes, unique diplotypes, and heterozygosity, as well as high average MAF, and low average pairwise Pearson correlation across all SNPs (average LD) (Spearman correlation $\rho \geq 0.22$, p≤$9.8 \times 10^{-04}$). Similarly, we quantified genomic segment complexity by the proportion of variance explained by the first two principal components as well as the number of principal components needed to explain at least 90% of the variance of HRC genotypes from each genomic segment. Concordantly, superior autoencoder performance was associated with a low proportion explained by the first two components and positively correlated with the number of components required to explained 90% of variance (Spearman $\rho \geq 0.22$, p≤$8.3 \times 10^{-04}$). These observations, with predictive features determined in the HRC training dataset and performance determined in the ARIC validation dataset, informed our tuning strategy.

We then used the genomic features significantly correlated with imputation performance in the ARIC validation dataset to predict the performance of and select the hyperparameter values to advance to fine-tuning. An ensemble model inference approach was able to predict the genomic segment-specific performance of hyperparameter combinations with high accuracy (*Figure 2—figure supplement 4*, mean r-squared=0.935 ± 0.002 of predicted vs observed autoencoders accuracies via 10-fold cross validation). The top 10 best performing hyperparameter combinations were advanced to fine-tuning (*Table 3*). Autoencoder tuning with simulated offspring formation was then executed as described in Materials and methods.

## Tuned performance

After tuning, autoencoder performance surpassed HMM-based imputation performance across all imputation methods, independent test datasets, and genotyping array marker sets. At a minimum, autoencoders surpassed HMM-based imputation performance in >62% of chromosome 22 genomic segments (two proportion test p=$1.02 \times 10^{-11}$) (Minimac4: *Figure 3*, Beagle5 and Eagle5: *Figure 3—figure supplements 1–2*). Overall, the optimized autoencoders exhibited superior performance with an average r-squared of 0.395±0.007 vs 0.374±0.007 for Minimac4 (Wilcoxon rank sum test p=0.007), 0.364±0.007 for Beagle5 (Wilcoxon rank sum test p=$1.53*10^{-4}$), and 0.358±0.007 for Impute5 (Wilcoxon rank sum test p=$2.01*10^{-5}$) (*Table 4*). This superiority was robust to the marker sets tested, with the mean r-squared per genomic segment for autoencoders being 0.373±0.008, 0.399±0.007, and 0.414±0.008 vs 0.352±0.008, 0.370±0.006, and 0.400±0.007 for Minimac4 using Affymetrix 6.0, UKB Axiom, and Omni 1.5 M marker sets (Wilcoxon rank-sums test *P*-value = 0.029, $1.99*10^{-4}$, and 0.087, respectively). Detailed comparisons to Beagle5 and Eagle5 are presented in *Figure 3—figure supplements 1–2*.

Tuning improved performance of the autoencoders across all genomic segments, generally improving the superiority of autoencoders relative to HMM-based approaches in genomic segments with complex haplotype structures while equalizing performance relative to HMM-based approaches in genomic segments with more simple LD structures (as described in Materials and methods, by the number of unique haplotypes: *Figure 3—figure supplement 3*, diplotypes: *Figure 3—figure supplement 4*, average pairwise LD: *Figure 3—figure supplement 5*, proportion variance explained: *Figure 3—figure supplement 6*). Concordantly, genomic segments with higher recombination rates exhibited the largest degree of improvement with tuning (*Figure 3—figure supplement 7*). Use of the augmented reference panel did not improve HMM-based imputation, having no influence on Minimac4 performance (original overall r-squared of 0.374±0.007 vs 0.363±0.007 after augmentation, Wilcoxon rank-sum test p=0.0917), and significantly degrading performance of Beagle5 and Impute5 (original r-squared of 0.364±0.007 and 0.358±0.007 vs 0.349±0.006 and 0.324±0.007 after augmentation, p=0.026 and p=$1.26*10^{-4}$, respectively). Summary statistics for these comparisons are available in *Supplementary file 1*.

## Overall chromosome 22 imputation accuracy

After merging the results from all genomic segments, the whole chromosome accuracy of autoencoder-based imputation remained superior to all HMM-based imputation tools, across all independent test datasets, and all genotyping array marker sets (Wilcoxon rank-sums test p≤$5.55 \times 10^{-67}$). The

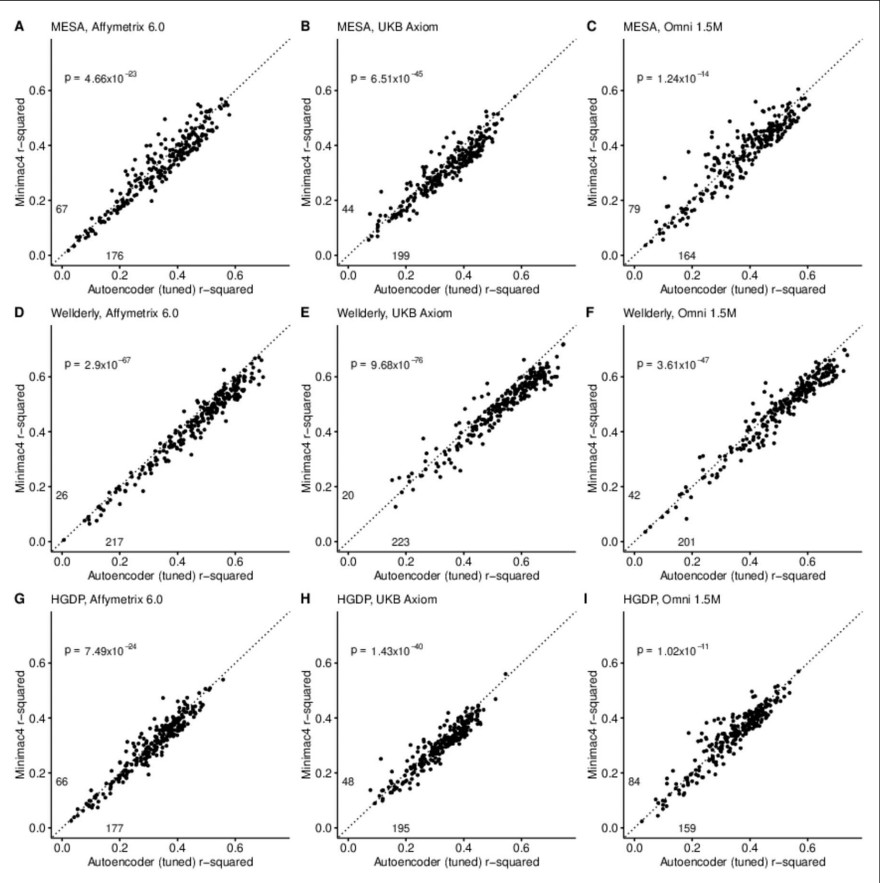

**Figure 3.** HMM-based (y-axis) versus autoencoder-based (axis) imputation accuracy after tuning. Minimac4 and tuned autoencoders were validated across three independent datasets–- MESA (top), Wellderly (**middle**), and HGDP (**bottom**) and across three genotyping array platforms–- Affymetrix 6.0 (**left**), UKB Axiom (**middle**), Omni1.5M (**right**). Each data point represents the imputation accuracy (average r-squared per variant) for an individual genomic segment relative to its WGS-based ground truth. The numerical values presented on the left side and below the identity line (dashed line) indicate the number of genomic segments in which Minimac4 outperformed the untuned autoencoder (left of identity line) and the number of genomic segments in which the untuned autoencoder surpassed Minimac4 (below the identity line). Statistical significance was assessed through two-proportion Z-test p-values.

The online version of this article includes the following figure supplement(s) for figure 3:

**Figure supplement 1.** Beagle5 (y-axis) versus autoencoder-based (axis) imputation accuracy after tuning.

**Figure supplement 2.** Impute5 (y-axis) versus autoencoder-based (axis) imputation accuracy after tuning.

**Figure supplement 3.** Imputation accuracy as a function of unique haplotype abundance.

**Figure supplement 4.** Imputation accuracy as a function of unique diplotype abundance.

**Figure supplement 5.** Imputation accuracy as a function of linkage disequilibrium (LD).

**Figure supplement 6.** Imputation accuracy as a function of data complexity.

**Figure supplement 7.** Imputation accuracy as a function of recombination rate.

---

autoencoder's mean r-squared per variant ranged from 0.363 for HGDP to 0.605 for the Wellderly vs 0.340–0.557 for Minimac4, 0.326–0.549 for Beagle5, and 0.314–0.547 for Eagle5, respectively. Detailed comparisons are presented in in *Table 5* and *Supplementary file 2*.

Further, when imputation accuracy is stratified by MAF bins, the autoencoders maintain superiority across all MAF bins by nearly all test dataset and genotyping array marker sets (*Figure 4*, and *Supplementary file 3*). Concordantly, autoencoder imputation accuracy is similarly superior when measured with F1-scores (*Figure 4—figure supplement 1*) and concordance (*Figure 4—figure supplement 2*), though these metrics are less sensitive at capturing differences in rare variant imputation accuracy.

**Table 4.** Performance comparisons between tuned autoencoder (AE) and HMM-based imputation tools (Minimac4, Beagle5, and Impute5).

Average r-squared per variant was extracted from each genomic segment of chromosome 22. We applied Wilcoxon rank-sum tests to compare the HMM-based tools to the reference untuned autoencoder (AE). * represents p-values ≤0.05, ** indicates p-values ≤0.001, and *** indicates p-values ≤0.0001.

| | MESA | Wellderly | HGDP | Affymetrix 6.0 | UKB Axiom | Omni 1.5 M | Combined |
|---|---|---|---|---|---|---|---|
| AE (tuned) | 0.355±0.007 | 0.505±0.008 | 0.327±0.006 | 0.373±0.008 | 0.399±0.007 | 0.414±0.008 | 0.396±0.007 |
| AE (untuned) | 0.303±0.008*** | 0.470±0.009* | 0.285±0.006*** | 0.339±0.008* | 0.356±0.007*** | 0.362±0.008*** | 0.352±0.008*** |
| Minimac4 | 0.337±0.007* | 0.471±0.008** | 0.314±0.006 | 0.352±0.008* | 0.370±0.006** | 0.400±0.007 | 0.374±0.007* |
| Beagle5 | 0.336±0.007* | 0.460±0.008*** | 0.296±0.005*** | 0.342±0.007** | 0.367±0.006*** | 0.384±0.007** | 0.364±0.007** |
| Impute5 | 0.326±0.007* | 0.458±0.008*** | 0.289±0.006*** | 0.336±0.008** | 0.354±0.006*** | 0.383±0.008** | 0.358±0.007*** |

When we upgraded the reference panel of the HMM-based tools with the more expansive TOPMed cohort, the superior performance of the HRC-trained autoencoder was still sustained across all datasets except for MESA (*Figure 4—figure supplement 3*). Given that MESA is a sub-cohort of the TOPMed cohort, we evaluated the possibility of residual data leakage after the removal of MESA from the TOPMed cohort and found that 44 MESA individuals were duplicated in other TOPMed cohorts, 182 MESA individuals had a first degree relative in other TOPMed cohorts, and >92% of MESA individuals had at least one second degree relative in other TOPMed cohorts, resulting in improved imputation performance. Notably, across the most diverse and truly independent HGDP validation dataset, the autoencoder displays superior performance despite only being exposed to training on the less diverse HRC reference cohort.

## Ancestry-specific chromosome 22 imputation accuracy

Finally, we evaluated ancestry-specific imputation accuracy. As before, overall autoencoder-based imputation maintains superiority across all continental populations present in MESA (*Figure 5*, Wilcoxon rank-sums test p=5.39 × 10$^{-19}$). The autoencoders' mean r-squared ranged from 0.357 for African ancestry to 0.614 for East Asian ancestry vs 0.328–0.593 for Minimac4, 0.330–0.544 for Beagle5, and 0.324–0.586 for Impute5, respectively. Note, East Asian ancestry exhibits a slightly higher overall imputation accuracy relative to European ancestry due to improved rare variant imputation. Autoencoder superiority replicates when HGDP is split into continental populations (*Figure 5—figure supplement 1*).

Further stratification of ancestry-specific imputation accuracy results by MAF continues to support autoencoder superiority across all ancestries, MAF bins, and nearly all test datasets, and genotyping array marker sets (*Figure 5*, *Figure 5—figure supplement 1*). Minimum and maximum accuracies across MAF by ancestry bins ranged between 0.177–0.937 for the autoencoder, 0.132–0.907 for Minimac4, 0.147–0.909 for Beagle5, and 0.115–0.903 for Impute5, with a maximum standard error of ±0.004.

**Table 5.** Whole chromosome level comparisons between autoencoder (AE) and HMM-based imputation tools (Minimac4, Beagle5, and Impute5).

Average r-squared per variant was extracted at whole chromosome level. We applied Wilcoxon rank-sum tests to compare the HMM-based tools to the reference tuned autoencoder (AE). * represents p-values ≤0.05, ** indicates p-values ≤0.001, and *** indicates p-values ≤0.0001. Standard errors that are equal or less than 0.001 are not shown.

| | MESA | | | Wellderly | | | HGDP | | |
|---|---|---|---|---|---|---|---|---|---|
| | Affymetrix 6.0 | UKB Axiom | Omni 1.5 M | Affymetrix 6.0 | UKB Axiom | Omni 1.5 M | Affymetrix 6.0 | UKB Axiom | Omni 1.5 M |
| AE (tuned) | 0.410 | 0.395 | 0.452 | 0.537 | 0.605 | 0.586 | 0.363 | 0.364 | 0.392 |
| Minimac4 | 0.390*** | 0.364*** | 0.436*** | 0.500*** | 0.557*** | 0.551*** | 0.350*** | 0.340*** | 0.385*** |
| Beagle5 | 0.383*** | 0.379*** | 0.420*** | 0.484*** | 0.549*** | 0.534*** | 0.326*** | 0.328*** | 0.353*** |
| Impute5 | 0.384*** | 0.356*** | 0.429*** | 0.485*** | 0.547*** | 0.539*** | 0.328*** | 0.314*** | 0.359*** |

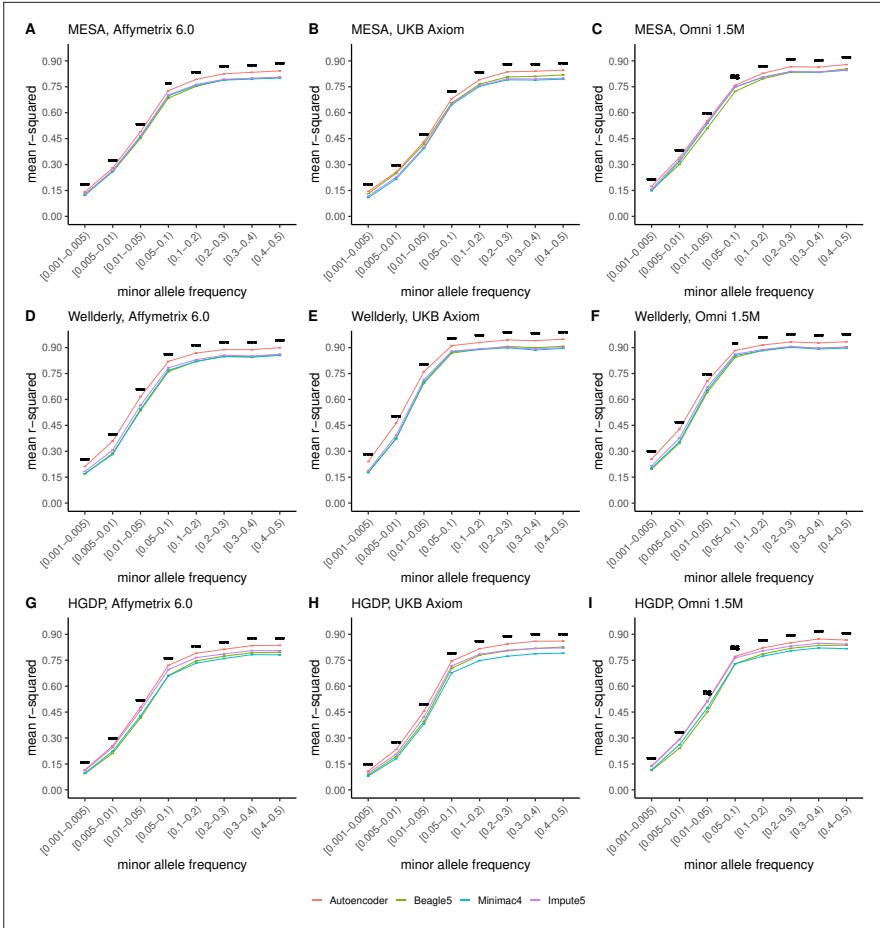

**Figure 4.** HMM-based versus autoencoder-based imputation accuracy across MAF bins. Autoencoder-based (**red**) and HMM-based (Minimac4 (**blue**), Beagle5 (**green**), and Impute5 (**purple**)) imputation accuracy was validated across three independent datasets–- MESA (**top**), Wellderly (**middle**), and HGDP (**bottom**) and across three genotyping array platforms–- Affymetrix 6.0 (**left**), UKB Axiom (**middle**), Omni1.5M (**right**). Each data point represents the imputation accuracy (average r-squared per variant) relative to WGS-based ground truth across MAF bins. Error bars represent standard errors. We applied Wilcoxon rank-sum tests to compare the HMM-based tools to the tuned autoencoder (AE). * represents p-values ≤0.05, ** indicates p-values ≤0.001, and *** indicates p-values ≤0.0001, ns represents non-significant p-values.

The online version of this article includes the following figure supplement(s) for figure 4:

**Figure supplement 1.** HMM-based versus autoencoder-based imputation accuracy across MAF bins (F1 score).

**Figure supplement 2.** HMM-based versus autoencoder-based imputation accuracy across MAF bins (concordance).

**Figure supplement 3.** TOPMed cohort HMM-based imputation versus HRC cohort autoencoder-based imputation accuracy across MAF bins.

Thus, with training on equivalent reference cohorts, autoencoder performance was superior across all variant allele frequencies and ancestries with the primary source of superiority arising from hard to impute regions with complex LD structures. When the reference panel of the HMM-based tools is upgraded to the more diverse TOPMed dataset, the HRC-trained autoencoder remains superior across all ancestry groups of HGDP (*Figure 5—figure supplement 2*), as well as in the MESA ancestries well represented in HRC (European and East Asian) but not in MESA ancestries where representation is significantly enhanced by the TOPMed reference panel (American and African) with additional imputation performance deriving from a significant degree of familial relationships spanning the TOPMed reference panel and MESA test cohort (*Figure 5—figure supplement 3*).

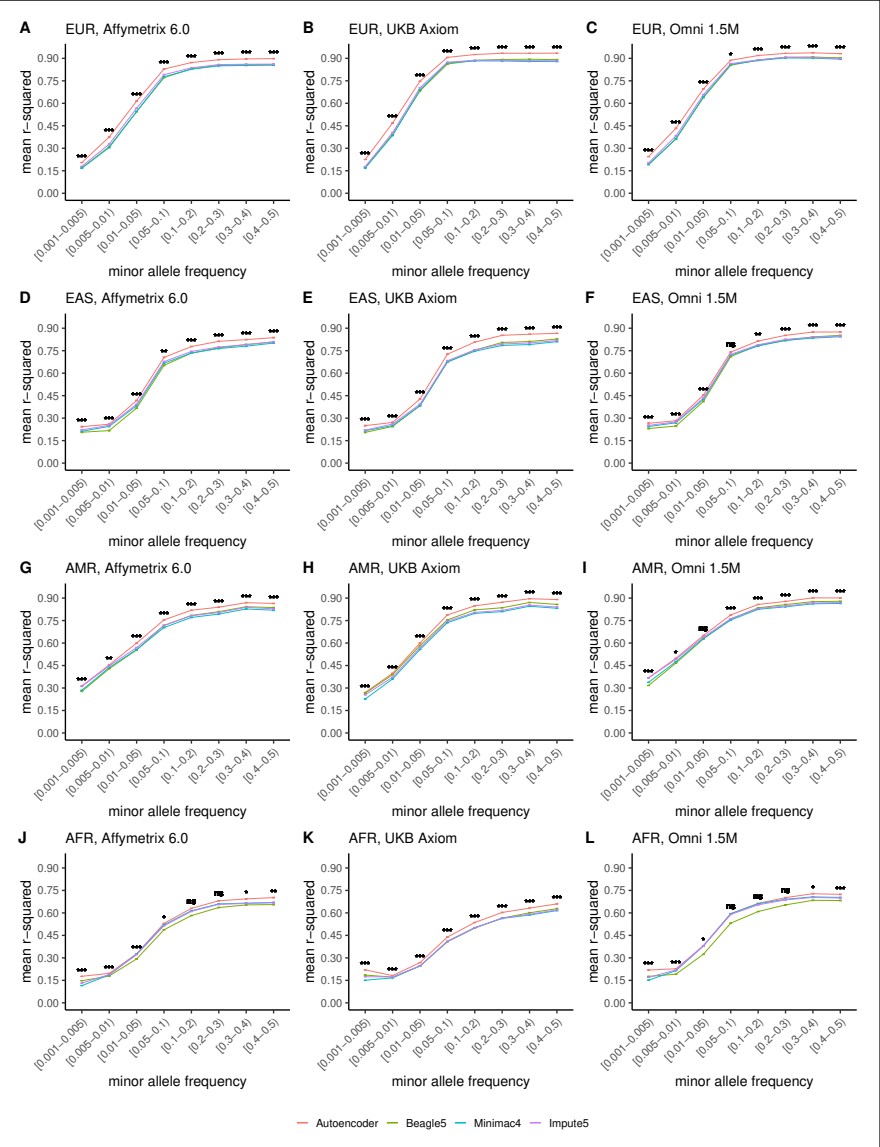

**Figure 5.** HMM-based versus autoencoder-based imputation accuracy across ancestry groups. Autoencoder-based (**red**) and HMM-based (Minimac4 (**blue**), Beagle5 (**green**), and Impute5 (**purple**)) imputation accuracy was validated across individuals of diverse ancestry from MESA cohort (EUR: European (**top**); EAS: East Asian (2nd **row**); AMR: Native American (3rd **row**); AFR: African (**bottom**)) and multiple genotype array platforms (Affymetrix 6.0 (**left**), UKB Axiom (**middle**), Omni1.5M (**right**)). Each data point represents the imputation accuracy (average r-squared per variant) relative to WGS-based ground truth across MAF bins. Error bars represent standard errors. We applied Wilcoxon rank-sum tests to compare the HMM-based tools to the tuned autoencoder (AE). * represents p-values ≤0.05, ** indicates p-values ≤0.001, and *** indicates p-values ≤0.0001, ns represents non-significant p-values.

The online version of this article includes the following figure supplement(s) for figure 5:

**Figure supplement 1.** HMM-based versus autoencoder-based imputation accuracy across ancestry groups.

**Figure supplement 2.** TOPMed cohort HMM-based versus HRC cohort autoencoder-based imputation accuracy across ancestry groups.

**Figure supplement 3.** TOPMed cohort HMM-based versus HRC cohort autoencoder-based imputation accuracy across ancestry groups.

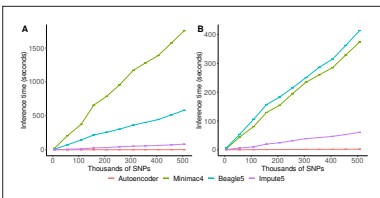

**Figure 6.** HMM-based versus autoencoder-based inference runtimes. We plot the average time and standard error of three imputation replicates. Two hardware configurations were used for the tests: (**A**) a low-end environment: 16-core Intel Xeon CPU (E5-2640 v2 2.00 GHz), 250 GB RAM, and one GPU (NVIDIA GTX 1080); (**B**) a high-end environment: 24-Core AMD CPU (EPYC 7352 2.3 GHz), 250 GB RAM, using one NVIDIA A100 GPU.

## Inference speed

Inference runtimes for the autoencoder vs HMM-based methods were compared in a low-end and high-end computational environment as described in *Methods*. In the low-end environment, the autoencoder's inference time is at least ~4 X faster than all HMM-based inference times (summing all inference times from all genomic segments of chromosome 22, the inference time for the autoencoder was $2.4 \pm 1.1 \times 10^{-3}$ seconds versus $1,754 \pm 3.2$, $583.3 \pm 0.01$, and $8.4 \pm 4.3 \times 10^{-3}$ s for Minimac4, Beagle5, and Impute5, respectively (*Figure 6A*)). In the high-end environment, this difference narrows to a~3 X advantage of the autoencoder vs HMM-based methods ($2.1 \pm 8.0 \times 10^{-4}$ versus $374.3 \pm 1.2$, $414.3 \pm 0.01$, and $6.1 \pm 2.1 \times 10^{-4}$) seconds for Minimac4, Beagle5, and Impute5, respectively (*Figure 6B*). These unoptimized results indicate that autoencoder-based imputation can be executed rapidly, without a reference cohort, and without the need for a high-end server or high-performance computing (HPC) infrastructure. However, we must note that to deploy the autoencoder-based imputation to production, the autoencoders must be pre-trained separately across all segments of all chromosomes in the human genome. This initial pre-training can require months of computation time, depending upon the GPU resources available, whereas HMM-based imputation does not require any pre-training after initial parameters are defined. Thus, the HMM-based approach is more flexible to the de-novo use of alternative reference panels – though recent cohorts have revealed scaling limitations. On the other hand, unlike HMM-based imputation tools, pre-trained autoencoders retain the information learned from pre-training and can be continuously fine-tuned with additional genomes and reference panels as they become available. Thus, once pre-trained, autoencoders may be incrementally upgraded using newly available reference panels.

## Discussion

Artificial neural network-based data mining techniques are revolutionizing biomedical informatics and analytics (*Dias and Torkamani, 2019*; *Jumper et al., 2021*). Here, we have demonstrated the potential for these techniques to execute a fundamental analytical task in population genetics, genotype imputation, producing superior results in a computational efficient and portable framework. The trained autoencoders can be transferred easily, and execute their functions rapidly, even in modest computing environments, obviating the need to transfer private genotype data to external imputation servers or services. Furthermore, our fully trained autoencoders robustly surpass the performance of all modern HMM-based imputation approaches across all tested independent datasets, genotyping array marker sets, minor allele frequency spectra, and diverse ancestry groups. This superiority was most apparent in genomic regions with low LD and/or high complexity in their linkage disequilibrium structure.

Superior imputation accuracy is expected to improve GWAS power, enable more complete coverage in meta-analyses, and improve causal variant identification through fine-mapping. Moreover, superior imputation accuracy in low LD regions may enable the more accurate interrogation of specific classes of genes under a greater degree of selective pressure and involved in environmental sensing. For example, promoter regions of genes associated with inflammatory immune responses, response to pathogens, environmental sensing, and neurophysiological processes (including sensory perception genes) are often located in regions of low LD (*Dias and Torkamani, 2019*; *Frazer et al., 2007*). These known disease-associated biological processes that are critical to interrogate accurately in GWAS. Thus, the autoencoder-based imputation approach both improves statistical power and biological coverage of individual GWAS' and downstream meta-analyses.

HMM-based imputation tools depend on end-user access to large reference panels or datasets to impute a single genome whereas pre-trained autoencoder models eliminate that dependency. However, further development is required to actualize this approach in practice for broad adoption. Autoencoders must be pre-trained and validated across all segments of the human genome – a computationally expensive task. Here we performed training only for chromosome 22. Autoencoder training is computationally intensive, shifting the computational burden to model trainers, and driving performance gains for end-users. As a result, inference time scales only with the number of variants to be imputed, whereas HMM-based inference time depends on both reference panel and the number of variants to be imputed. This allows for autoencoder-based imputation to extend to millions of genomes but introduces some challenges in the continuous re-training and fine-tuning of the pre-trained models as larger reference panels are made available. In addition, our current encoding approach lacks phasing information and no pre-phasing was performed. Pre-phasing can lead to substantial improvements in imputation accuracy. Future models will need to address the need for phasing and continuous fine-tuning of models for application to modern, ever-growing, genomic datasets.

## Ideas and speculation

After expanding this approach across the whole genome, our work will provide a more efficient genotype imputation platform on whole genome scale and thus benefit genomic research especially in contexts where the computational power required for modern HMM-based imputation is not accessible. In addition to the speed, cost, and accuracy benefits, our proposed approach can potentially improve automation for downstream analyses. The autoencoder naturally generates a hidden encoding with latent features representative of the original data. This latent representation of the original data acts as an automatic feature extraction and dimensionality reduction technique for downstream tasks such as genetic risk prediction. Moreover, the autoencoder-based imputation approach only requires a reference panel during training – only the neural network needs to be distributed for implementation. Thus, the neural network is portable and avoids privacy issues associated with standard statistical imputation. This privacy-preserving feature will allow developers to deploy real-time data-driven algorithms on personal devices (edge computing). These new features will expand the clinical applications of genomic imputation, as well as its role in preventive healthcare. Another point related to data privacy is that the autoencoders segment the genome, making reconstruction of an individual genome impossible even if reference data were somehow recoverable from the neural networks. Nevertheless, while there are no official data sharing restrictions on deep learning model weights generated from genomic data, future privacy risks may be discovered, necessitating further research into privacy concerns and differential privacy techniques for autoencoder-based genotype imputation.

## Acknowledgements

This work is supported by R01HG010881 to AT, KL2TR002552 to RD, as well as grants U24TR002306 and UL1TR002550. We would like to thank JC Ducom and Lisa Dong from the Scripps High Performance Computing center, as well as Fernanda Foertter, Johnny Israeli, Ohad Mosafi, and Joyjit Daw from NVIDIA for their technical support and collaboration in this project. A portion of this research was conducted using a startup account at the Summit supercomputer from Oak Ridge National Laboratory (ORNL).

## Additional information

### Funding

| Funder | Grant reference number | Author |
|---|---|---|
| National Institutes of Health | R01HG010881 | Raquel Dias |
| National Institutes of Health | KL2TR002552 | Raquel Dias |

| Funder | Grant reference number | Author |
| --- | --- | --- |
| National Institutes of Health | U24TR002306 | Doug Evans |
| National Institutes of Health | UL1TR002550 | Doug Evans |

The funders had no role in study design, data collection and interpretation, or the decision to submit the work for publication.

### Author contributions

Raquel Dias, Conceptualization, Data curation, Software, Formal analysis, Validation, Investigation, Visualization, Methodology, Writing – original draft, Writing – review and editing; Doug Evans, Data curation, Software, Visualization, Methodology, Writing – review and editing; Shang-Fu Chen, Data curation, Software, Validation, Methodology; Kai-Yu Chen, Data curation, Software, Validation; Salvatore Loguercio, Data curation, Formal analysis, Validation, Visualization, Writing – review and editing; Leslie Chan, Software, Validation, Methodology; Ali Torkamani, Conceptualization, Resources, Supervision, Funding acquisition, Investigation, Project administration, Writing – review and editing

### Author ORCIDs

Ali Torkamani http://orcid.org/0000-0003-0232-8053

### Decision letter and Author response

Decision letter https://doi.org/10.7554/eLife.75600.sa1
Author response https://doi.org/10.7554/eLife.75600.sa2

## Additional files

### Supplementary files

• Transparent reporting form

• Supplementary file 1. Performance comparisons between tuned autoencoder (AE) and HMM-based imputation tools (Minimac4, Beagle5, and Impute5) after applying data augmentation to HMM-based tools.

• Supplementary file 2. Detailed performance comparisons between tuned autoencoder (AE) and HMM-based imputation tools (Minimac4, Beagle5, and Impute5).

• Supplementary file 3. Detailed performance comparisons between tuned autoencoder (AE) and HMM-based imputation tools (Minimac4, Beagle5, and Impute5).

### Data availability

The data that support the findings of this study are available from dbGAP and European Genome-phenome Archive (EGA), but restrictions apply to the availability of these data, which were used under ethics approval for the current study, and so are not openly available to the public. The computational pipeline for autoencoder training and validation is available at https://github.com/TorkamaniLab/Imputation_Autoencoder/tree/master/autoencoder_tuning_pipeline (copy archived at swh:1:rev:35d2e292e786ebc41e71f27809dad56b1e1933c4). The python script for calculating imputation accuracy is available at https://github.com/TorkamaniLab/imputation_accuracy_calculator, (copy archived at swh:1:rev:e01229e3f245e8bb95b29d4f4f1e547fcff70ae4). Instructions on how to access the unique information on the parameters and hyperparameters of each one of the 256 autoencoders is shared through our source code repository at https://github.com/TorkamaniLab/imputator_inference, (copy archived at swh:1:rev:2fbd203acf8aaf320a520c6374d6f4d57f068a7c). We also shared the pre-trained autoencoders and instructions on how to use them for imputation at https://github.com/TorkamaniLab/imputator_inference. Imputation data format. The imputation results are exported in variant calling format (VCF) containing the imputed genotypes and imputation quality scores in the form of class probabilities for each one of the three possible genotypes (homozygous reference, heterozygous, and homozygous alternate allele). The probabilities can be used for quality control of the imputation results.

The following previously published datasets were used:

| Author(s) | Year | Dataset title | Dataset URL | Database and Identifier |
|---|---|---|---|---|
| McCarthy S, Das S, Kretzschmar W | 2016 | Haplotype Reference Consortium | https://ega-archive.org/studies/EGAS00001001710 | EGA European Genome-Phenome Archive, EGAS00001001710 |
| 1000 Genomes Project Consortium | 2015 | The 1000 Genomes Project Consortium | https://www.internationalgenome.org/data-portal/data-collection/phase-3 | Internationalgenome, phase-3 |
| Bild DE, Bluemke DA, Burke GL | 2002 | MESA (Multi-Ethnic Study of Atherosclerosis) study | https://www.ncbi.nlm.nih.gov/projects/gap/cgi-bin/study.cgi?study_id=phs001416.v2.p1 | NCBI Gene Expression Omnibus, phs001416.v2.p1 |
| The ARIC investigators consortium | 1989 | Atherosclerosis Risk in Communities (ARIC) | https://www.ncbi.nlm.nih.gov/projects/gap/cgi-bin/study.cgi?study_id=phs001211.v4.p3 | NCBI Gene Expression Omnibus, phs001211.v4.p3 |
| Bergström A, McCarthy SA, Hui R | 2020 | Human Genome Diversity Project (HGDP) | https://www.internationalgenome.org/data-portal/data-collection/hgdp | Internationalgenome, hgdp |
| Taliun D, Harris DN, Kessler MD | 2021 | TOPMed Cohort | https://topmed.nhlbi.nih.gov/ | Multiple, topmed |

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
