## [Editor Report]

The paper describes a novel neural-network-based strategy for imputing unmeasured genotypes, which is a standard part of most association testing pipelines. The method is computationally intensive to train, but once training is complete the imputation is fast and accurate and does not require further access to a reference panel. It has the potential to be a practically-appealing alternative to existing methods. although further work (eg training of models) is required before this new approach can be applied genome-wide.

---

## [Decision Letter]

**Decision letter after peer review:**

Thank you for submitting your article "Rapid, Reference-Free Human Genotype Imputation with Denoising Autoencoders" for consideration by *eLife*. Your article has been reviewed by 4 peer reviewers, one of whom is a member of our Board of Reviewing Editors, and the evaluation has been overseen by Molly Przeworski as the Senior Editor. The reviewers have opted to remain anonymous.

Essential revisions:

1) More precise technical details need to be provided about the method. Code used to train the method should be made available.

2) Accuracy assessments should be made on a new dataset not used at any point in the training.

3) Comparison with HMM should use a more diverse reference panel and with standard quality controls in place, with separate comparisons for rare vs common variants.

4) The reason for the surprisingly low average accuracy of all methods, compared with

previous studies, needs to be identified and, probably, corrected. (Reported imputation accuracies in previous studies are typically R2>0.8, whereas in this paper reported R2 is mostly in range 0.2-0.6).

5) Software for applying the pre-trained network to impute genotypes needs to be publicly available.

*Reviewer #1 (Recommendations for the authors):*

Supplemental Figures 12 and 13 – I appreciate the authors using various metrics for comparing imputation accuracies, but I think these figures are a bit confusing because it shows that SNPs with smaller MAF somehow have better imputation accuracy.

*Reviewer #2 (Recommendations for the authors):*

Two particular concerns for the authors:

1) The code is not yet available. Once it is made available I'll be able to more thoroughly evaluate what the authors have done.

2) As mentioned above, there is a bit of work to do on cleaning up the run time performance evaluations.

*Reviewer #3 (Recommendations for the authors):*

Besides the points mentioned in the public review, I have the following comments/suggestions:

– While I am somewhat familiar with autoencoders, I found it quite hard to follow the description of "denoising" autoencoders. Part of the problem was that I did not appreciate that these are different from a regular autoencoder until I did a bit more reading. I'm not quite sure how to make the description more accessible to the audience who are not familiar with denoising autoencoders, but I think it would have helped me to have an equation giving the training objective function that is being optimized to supplement Figure 1.

– The description of the HMM methods on p3 is inaccurate. For example, the hidden states are not the "to-be-imputed" variants (a better high-level description would be that the hidden states represent the haplotype in the reference panel that is most closely related to the haplotype being imputed, although different HMMs may have slightly different interpretations for the hidden states) and the genotyped variants are not the "observed states" (they are observed, but they are not the HMM states). In any case it needs rewriting.

– p4 the analogy with single-cell "dropout" is misleading because the zeros in single cell data are not "missing data" in the same sense as the missing genotypes in genotype imputation. (See https://www.nature.com/articles/s41588-021-00873-4 for example).

– p6: although ultimately performance is what matters, the genotype encoding seems a bit weird to me: specifically the way that 0 is used for both missing data and absence of the allele (which are different things!) Is this standard in the denoising autoencoder literature? Also, can you clarify what you mean by "scaled outputs can also be regarded as probabilities" and "This representation maintains the interdepencies among classes". Indeed, I don't see why this last is true: for example, the "true" genotype can never be (0,0), so if one of the pair is 0 the other must be 1, and this interdependency does not seem to be captured by the encoding.

*Reviewer #4 (Recommendations for the authors):*

Specific comments are given below.

1. The authors trained their models using HRC reference panel, which consists of predominantly European ancestry individuals. They didn't mention the reference panel used in HMM-based imputations, and I assume it was also HRC. As we know, performance of HMM-based imputation depends on the reference panels. There are larger and more diverse reference panels publicly available (e.g. TOPMed freeze 8 reference panel through the TOPMed imputation server). Does AE still have values (whatever training datasets they are using) over the TOPMed imputation server? Evaluations with diverse reference panel is particularly important for at least two reasons: (1) diverse panels are widely used (if not the most widely used); (2) AE relies more on pretrained model that no longer retains individual haplotype level information than HMM-based methods, which is conceptually susceptible to heterogeneity in reference panel. The authors may not have access to individual-level TOPMed data for training their AE models. In that case, they should at least evaluate the performance with the 1000 Genomes Project as reference.

2. The authors didn't mention any post-imputation QC steps, thus it's not clear to me which variants are included in their evaluations/comparisons. As we know, not all markers attempted (that is, markers in the reference panel) can be well imputed. It is essential to have post-imputation QC to filter out poorly imputed markers. The manuscript does not touch this important topic at all, rendering the method practically not useful. Also rather puzzlingly, the R2 showed in all the figures are largely in the range 0.2-0.6, which is inconsistent from what's reported in the literature. For common variants, HMM-based imputation methods have been reported to attain R2 > 0.8 (if not even much higher); and for rare variants, the mode is close to zero. The expected bimodal distribution is not observed. Please clarify. For similar reason, quality assessment should be performed for common and rare variants separately.

3. Have the authors evaluated the computationally identified hotspots? Are they largely consistent with the "true" recombination hotspots? Based on Figure 1E, these hotspots are calculated according to LD. What if the reference panel contains diverse populations? The LD calculated based on heterogeneous individuals would be meaningless.

[Editors’ note: further revisions were suggested prior to acceptance, as described below.]

Thank you for resubmitting your work entitled "Rapid, Reference-Free Human Genotype Imputation with Denoising Autoencoders" for further consideration by *eLife*. Your revised article has been evaluated by two of the original Reviewers, Molly Przeworski (Senior Editor) and a Reviewing Editor.

The manuscript has been improved but there are some remaining issues that need to be addressed, as outlined below:

1. You need to clearly report the (very considerable) training time required for the autoencoder. This relates to comments from both Reviewers which I repeat here:

Reviewer 2: "I don't think the authors are dealing enough with performance issues and the need to share training times for tuning. First, in the revision the authors have edited/added Figure 6 which shows a runtime comparison, however, the methods section description (lines 308-315) does not mention how only prediction time was taken into account, as the authors seem to be explaining in the Response to Reviews document. More importantly, given that software release is mostly a proof of principle- pre-trained models are available for chr22 only -further, genome-wide inference a user would have to (as I understand it): download a reference set, train tiled auto-encoders for the genome on their own hardware, and then do inference on their data of interest. Surely then it's important to profile the performance of this bit of the pipeline for the user."

Reviewer 4. That "the training process is going to take approximately one year to finish." would severely limit the application of the method. If the authors, as developers, do not release pre-trained models, we would be relying on extremely motivated and computationally savvy users to perform the pre-training. Along the line, reporting only the inference speed (i.e., computing time needed to APPLY the pre-trained models) becomes insufficient since the users would need to pre-train the models for all other chromosomes before they can apply the models to their target data.

2. You need to address the original request to compare against HMMs from a diverse reference. This request was made in the original decision, but the response instead focussed on the diversity of the target samples, which is a different issue. (It seems that running the autoencoder with a new diverse reference sample would require considerable new computation, but running the HMMs with a more diverse reference should be relatively straightforward; this does seem to highlight a disadvantage of the long training time of autoencoders if they are to be retrained when new larger reference samples become available.)

This relates to the following comment from Reviewer 4:

For major comment 1, I was not referring to a diverse target but referred to a diverse reference. I know that HRC includes the samples from the 1000 Genomes Project but the aggressive variant filtering and the predominant European ancestry individuals (~30,000 versus ~2,000 non-European from the 1000 Genomes Project) make it a lousy example as a diverse reference.

---

## [Author Response]

Essential revisions:1) More precise technical details need to be provided about the method. Code used to train the method should be made available.

We have made the source code and detailed instructions for use available publicly at Github.

The computational pipeline for autoencoder training and validation is available at:

https://github.com/TorkamaniLab/Imputation_Autoencoder/tree/master/autoencoder_tuning_pipeline.

The python script for calculating imputation accuracy is available at: https://github.com/TorkamaniLab/imputation_accuracy_calculator.

2) Accuracy assessments should be made on a new dataset not used at any point in the training.

The MESA, Wellderly, and HGDP datasets are all independent datasets, never used for training, nor model selection. Only HRC was used as reference panel or for training, and ARIC was used for model tuning. We included a statement in the methods clarifying this point.

3) Comparison with HMM should use a more diverse reference panel and with standard quality controls in place, with separate comparisons for rare vs common variants.

Two of our independent testing datasets (MESA, but especially HGDP), are from highly diversified multi-ethnic cohorts. HGDP includes 828 samples from 54 different populations representing all continental populations and including remote populations like Siberia, Oceania, etc. This reference panel is described in more detail in the reference below and likely represents the most diverse human genome dataset available.

Bergström A, et al., Insights into human genetic variation and population history from 929 diverse genomes. Science. 2020 Mar 20;367(6484):eaay5012.

4) The reason for the surprisingly low average accuracy of all methods, compared withprevious studies, needs to be identified and, probably, corrected. (Reported imputation accuracies in previous studies are typically R2>0.8, whereas in this paper reported R2 is mostly in range 0.2-0.6).

We show average accuracy of 0.2-0.6 in Table 4, but that is the average aggregate R2 per variant across all variants (no minor allele frequency filter or binning applied). The reviewer points that the accuracy should be R2>0.8, but this R2>0.8 expectation applies only to common variants (allele frequency >1%). Our results reflect this level of accuracy (R2>0.8) for common variants as plotted in Figure 4. The aggregate accuracy we report in Table 4 is substantially lower because the vast majority of genetic variants are rare, falling below the 1% allele frequency threshold and pulling down the overall average. The parent HRC reference publication below, as well as other supporting references we provide, demonstrate this is expected and conforms with our results.

References:

Rubinacci S, Delaneau O, Marchini J. Genotype imputation using the positional burrows wheeler transform. PLoS genetics. 2020 Nov 16;16(11):e1009049.

Our curves in Figure 4 replicate this graded accuracy across allele frequencies with higher accruacy above 1% allele frequency. Other examples:

McCarthy S, Das S, Kretzschmar W, Delaneau O, Wood AR, Teumer A, Kang HM, Fuchsberger C, Danecek P, Sharp K, Luo Y. A reference panel of 64,976 haplotypes for genotype imputation. Nature genetics. 2016 Oct;48(10):1279.

Vergara C, Parker MM, Franco L, Cho MH, Valencia-Duarte AV, Beaty TH, Duggal P. Genotype imputation performance of three reference panels using African ancestry individuals. Human genetics. 2018 Apr;137(4):281-92.

5) Software for applying the pre-trained network to impute genotypes needs to be publicly available.

We now share the pre-trained autoencoders (including model weights and inference source code), as well as instructions on how to use them for imputation.

These resources are publicly available at https://github.com/TorkamaniLab/imputator_inference. We have added this information to the Data Availability subsection of the Methods.

Reviewer #1 (Recommendations for the authors):Supplemental Figures 12 and 13 – I appreciate the authors using various metrics for comparing imputation accuracies, but I think these figures are a bit confusing because it shows that SNPs with smaller MAF somehow have better imputation accuracy.

Yes, this is a bit of an accuracy paradox: since most of the genotypes in rare variants are homozygous reference, concordances and F-scores are inflated due to the overwhelming abundance of the negative class. In order to address this source of confusion, we have added a note to the captions of these figures explaining this phenomenon and pointing the reader to Figure S14 (and R-squared in general) for a more accurate picture of balanced accuracy.

Reviewer #2 (Recommendations for the authors):Two particular concerns for the authors:1) The code is not yet available. Once it is made available I'll be able to more thoroughly evaluate what the authors have done.

We made both pre-trained models and the source code available now.

2) As mentioned above, there is a bit of work to do on cleaning up the run time performance evaluations.

We have clarified the major concerns on the performance comparisons, including further clarification that all the testing results shown are from independent datasets not used for training, nor tuning; and that the runtime comparisons extracted from HMM refer to the prediction step only, excluding the time for I/O and HMM iterations.

Reviewer #3 (Recommendations for the authors):Besides the points mentioned in the public review, I have the following comments/suggestions:– While I am somewhat familiar with autoencoders, I found it quite hard to follow the description of "denoising" autoencoders. Part of the problem was that I did not appreciate that these are different from a regular autoencoder until I did a bit more reading. I'm not quite sure how to make the description more accessible to the audience who are not familiar with denoising autoencoders, but I think it would have helped me to have an equation giving the training objective function that is being optimized to supplement Figure 1.

We edited/included a couple of sentences in the introduction and methods, defining what is the most basic difference between typical autoencoders and the denoising autoencoder. We hope this helps to further clarify this issue.

– The description of the HMM methods on p3 is inaccurate. For example, the hidden states are not the "to-be-imputed" variants (a better high-level description would be that the hidden states represent the haplotype in the reference panel that is most closely related to the haplotype being imputed, although different HMMs may have slightly different interpretations for the hidden states) and the genotyped variants are not the "observed states" (they are observed, but they are not the HMM states). In any case it needs rewriting.

Thanks for catching this. We corrected the HMM definition in the introduction.

– p4 the analogy with single-cell "dropout" is misleading because the zeros in single cell data are not "missing data" in the same sense as the missing genotypes in genotype imputation. (See https://www.nature.com/articles/s41588-021-00873-4 for example).

While there is some subtlety to this point, single-cell “dropout” includes both molecules that actually do not exist in the cell at the time of sampling as well as molecules that do exist but are not captured by the single-cell assay. Our own experience with scRNAseq data confirms this, taking the same library and sequencing it more deeply (or applying other molecular techniques to deplete abundant species) leads to the discovery of a larger number of genes per cell.

These are true missing data.

Despite this, we agree the scRNAseq example is complex and perhaps confusing – a simpler example more true to the nature of missing data in genotype data would be more appropriate. Therefore we’ve replaced those examples with the more classic image applications.

– p6: although ultimately performance is what matters, the genotype encoding seems a bit weird to me: specifically the way that 0 is used for both missing data and absence of the allele (which are different things!) Is this standard in the denoising autoencoder literature? Also, can you clarify what you mean by "scaled outputs can also be regarded as probabilities" and "This representation maintains the interdepencies among classes". Indeed, I don't see why this last is true: for example, the "true" genotype can never be (0,0), so if one of the pair is 0 the other must be 1, and this interdependency does not seem to be captured by the encoding.

We rescale the outputs by the following formula:

AE output example [x’ref, x’alt]: [0.55,0.90]

x′[ref,ref] = x′ref ∗ (1 − x′alt) = 0.55 ∗ 0.10 = 0.055

x′[alt,alt] = (1 − x′ref) ∗ x′alt = 0.45 ∗ 0.90 = 0.405

x′[ref,alt] = x′ref ∗ x′alt = 0.55 ∗ 0.90 = 0.495

where x’ is the reconstructed outputs from the autoencoders for reference (ref) and alternate (alt) alleles.

Next, we convert the rescaled outputs into probabilities ( P[ref,ref], P[alt,alt], P[alt,alt] ) by using a softmax function:P[ref ,ref]=ex′[ref ,ref]ex′[ref ,ref]+ex′[ref ,alt]+ex′[alt.alt]=0.252P[ref,alt]=ex′[ref,alt]ex′[ref,ref]+ex′[ref,alt]+ex′[alt,alt]=0.357P[alt,alt]=ex′[ref,alt]ex′[ref,ref]+ex′[ref,alt]+ex′[alt,alt]=0.391 Corrupting the data can be done by using any randomly chosen value as replacement for the true data. Either 0.0 or 0.5 is typically utilized for representing fixed noise or masking noise. Even in an image, RGB 0, 0, 0 is often used but is not actually “missing” but black. True “missing” is not used in autoencoders. We didn’t find any significant differences by using either of these values for masking.

We have removed the statement about interdependencies to avoid confusion. However, as a point of interest for the reviewer – while we have yet to test this point – 0,0 genotypes may actually be possible – when the position containing the SNP locus is deleted in the genome. This gets a bit complex and we’re not absolutely sure whether we would be able to capture this sort of event with our encoding, so we’ve simply removed the statement.

Reviewer #4 (Recommendations for the authors):Specific comments are given below.1. The authors trained their models using HRC reference panel, which consists of predominantly European ancestry individuals. They didn't mention the reference panel used in HMM-based imputations, and I assume it was also HRC. As we know, performance of HMM-based imputation depends on the reference panels. There are larger and more diverse reference panels publicly available (e.g. TOPMed freeze 8 reference panel through the TOPMed imputation server). Does AE still have values (whatever training datasets they are using) over the TOPMed imputation server?

We included further clarification in the methods section:

“We utilized HRC as reference panel for the HMM-based imputation tools, which is the same dataset used for training the autoencoders, and we applied the same quality control standards for both HMM-based and autoencoder-based imputation.”

Until recently, HRC has been the standard for imputation (and is still often used). Both the HMM-method and our autoencoders would improve with the use of an expanded reference. The fairest comparison is to use the same reference for both approaches.

Evaluations with diverse reference panel is particularly important for at least two reasons: (1) diverse panels are widely used (if not the most widely used); (2) AE relies more on pretrained model that no longer retains individual haplotype level information than HMM-based methods, which is conceptually susceptible to heterogeneity in reference panel. The authors may not have access to individual-level TOPMed data for training their AE models. In that case, they should at least evaluate the performance with the 1000 Genomes Project as reference.

We have evaluated our models on data that is significantly more diverse than the 1000 Genomes (note that 1000 Genomes is also not independent from HRC). Two of our independent testing datasets (MESA, but especially HGDP), are from highly diversified multi-ethnic cohorts. HGDP includes 828 samples from 54 different populations representing all continental populations and including remote populations like Siberia, Oceania, etc. This reference panel is described in more detail in the reference below and likely represents the most diverse human genome dataset available – more diverse (though smaller) than TopMed.

Bergström A, et al., Insights into human genetic variation and population history from 929 diverse genomes. Science. 2020 Mar 20;367(6484):eaay5012.

2. The authors didn't mention any post-imputation QC steps, thus it's not clear to me which variants are included in their evaluations/comparisons. As we know, not all markers attempted (that is, markers in the reference panel) can be well imputed. It is essential to have post-imputation QC to filter out poorly imputed markers. The manuscript does not touch this important topic at all, rendering the method practically not useful. Also rather puzzlingly, the R2 showed in all the figures are largely in the range 0.2-0.6, which is inconsistent from what's reported in the literature. For common variants, HMM-based imputation methods have been reported to attain R2 > 0.8 (if not even much higher); and for rare variants, the mode is close to zero. The expected bimodal distribution is not observed. Please clarify. For similar reason, quality assessment should be performed for common and rare variants separately.

We included further clarification in the methods explaining that no post-imputation QC was applied. However, we bin performance by allele frequency which relates to imputability. And the output of our model includes genotype probabilities which can be used as a measure of post-imputation QC for end-users. This point is now explained in the Genotype Encoding section of Methods, and in the documentation of the imputation software repository available at:

https://github.com/TorkamaniLab/imputator_inference

As for overall accuracy, we show average accuracy of 0.2-0.6 in Table 4, but that is the average aggregate R2 per variant across all variants (no MAF filter or binning applied). The reviewer points that the accuracy should be R2>0.8, but this R2>0.8 expectation applies only to common variants (allele frequency >1%). Our results reflect this level of accuracy (r2>0.8) for common variants as plotted in Figure 4. The aggregate accuracy we report in Table 4 is substantially lower because the vast majority of genetic variants are rare, falling below the 1% allele frequency threshold and pulling down the average. The parent HRC reference, as well as other supporting references we provide, demonstrate this is expected and conforms with our results.

References:

Rubinacci S, Delaneau O, Marchini J. Genotype imputation using the positional burrows wheeler transform. PLoS genetics. 2020 Nov 16;16(11):e1009049.

McCarthy S, Das S, Kretzschmar W, Delaneau O, Wood AR, Teumer A, Kang HM, Fuchsberger C, Danecek P, Sharp K, Luo Y. A reference panel of 64,976 haplotypes for genotype imputation. Nature genetics. 2016 Oct;48(10):1279.

Vergara C, Parker MM, Franco L, Cho MH, Valencia-Duarte AV, Beaty TH, Duggal P. Genotype imputation performance of three reference panels using African ancestry individuals. Human genetics. 2018 Apr;137(4):281-92.

3. Have the authors evaluated the computationally identified hotspots? Are they largely consistent with the "true" recombination hotspots? Based on Figure 1E, these hotspots are calculated according to LD. What if the reference panel contains diverse populations? The LD calculated based on heterogeneous individuals would be meaningless.

Although we have not manually verified this across the whole genome, we have done several comparisons across regions of known recombination hotspots versus our LD based tiling method. One of the examples is shown in Author response image 1, where the LD-based tile boundaries identified by our method are defined by the vertical red lines, and the recombination hotspots defined by PRDM9 binding sites are represented in green.

**Author response image 1. sa2fig1:** 

Similarly, we have compared our LD-based tiling boundaries of HRC versus 1000G. The plot (Author response image 2) shows an example where the HRC scores and 1000G scores are plotted and overlaid for some example regions from chromosome 10. The black line is HRC LD scores, the brown plot is the 1000G LD scores. The blue and red vertical lines are the genomic segment boundaries found by our method.

However, we had to further split certain segments that were too large due to GPU memory limitations (see Methods).As expected, the sub-segments resulting from this further split did not match to known recombination sites.

[Editors’ note: further revisions were suggested prior to acceptance, as described below.]

The manuscript has been improved but there are some remaining issues that need to be addressed, as outlined below:1. You need to clearly report the (very considerable) training time required for the autoencoder. This relates to comments from both Reviewers which I repeat here:Reviewer 2: "I don't think the authors are dealing enough with performance issues and the need to share training times for tuning. First, in the revision the authors have edited/added Figure 6 which shows a runtime comparison, however, the methods section description (lines 308-315) does not mention how only prediction time was taken into account, as the authors seem to be explaining in the Response to Reviews document. More importantly, given that software release is mostly a proof of principle- pre-trained models are available for chr22 only -further, genome-wide inference a user would have to (as I understand it): download a reference set, train tiled auto-encoders for the genome on their own hardware, and then do inference on their data of interest. Surely then it's important to profile the performance of this bit of the pipeline for the user."Reviewer 4. That "the training process is going to take approximately one year to finish." would severely limit the application of the method. If the authors, as developers, do not release pre-trained models, we would be relying on extremely motivated and computationally savvy users to perform the pre-training. Along the line, reporting only the inference speed (i.e., computing time needed to APPLY the pre-trained models) becomes insufficient since the users would need to pre-train the models for all other chromosomes before they can apply the models to their target data.

We have now expanded the description of the implications of the inference time at the end of the Results section with a discussion of the considerable training time required to generate pre-trained models that achieve our reported inference time. We contrast this with HMM-based imputation tools, statin that they are more easily adaptable to the de-novo application of novel reference panels (lines 541 – 551).

2. You need to address the original request to compare against HMMs from a diverse reference. This request was made in the original decision, but the response instead focussed on the diversity of the target samples, which is a different issue. (It seems that running the autoencoder with a new diverse reference sample would require considerable new computation, but running the HMMs with a more diverse reference should be relatively straightforward; this does seem to highlight a disadvantage of the long training time of autoencoders if they are to be retrained when new larger reference samples become available.)This relates to the following comment from Reviewer 4:For major comment 1, I was not referring to a diverse target but referred to a diverse reference. I know that HRC includes the samples from the 1000 Genomes Project but the aggressive variant filtering and the predominant European ancestry individuals (~30,000 versus ~2,000 non-European from the 1000 Genomes Project) make it a lousy example as a diverse reference.

We have acquired the TOPMed reference cohort and performed HMM-based imputation using this reference panel. A description of this reference panel is provided in methods (lines 147 – 160). Imputation accuracy comparisons are provided at the cohort level (line 476 – 485; new Figure 4—figure supplement 3) and broken down by ancestry (lines 521 – 528; new Figure 5—figure supplement 2 and 3). The autoencoder retains superior performance even when the HMM has access to a more diverse cohort except in the case of MESA – which is a component of TOPMed and contains a significant rate of familial relationship to the rest of the TOPMed cohort. Additional minor edits to support the addition of this analysis are made throughout the manuscript.